# Kiwifruit Adaptation to Rising Vapor Pressure Deficit Increases the Risk of Kiwifruit Decline Syndrome Occurrence

**Laura Bardi** [1,*] , **Luca Nari** [2] , **Chiara Morone** [3] , **Mauro Solomita** [1] , **Claudio Mandalà** [1] , **Maria Giulia Faga** [4]
and **Carmela Anna Migliori** [1]

1   Research Centre for Engineering and Agro-Food Processing, CREA Council for Agricultural Research and Economics, 10135 Turin, Italy
2   AGRION, The Foundation for Research, Innovation and Technological Development of Piedmont Agriculture, 12030 Manta, Italy
3   Phytosanitary and Scientific-Technical Services Department, Agricultural and Food Directorate, Piedmont Region, 10144 Turin, Italy
4   Italian National Research Council (CNR), Institute of Sciences and Technologies for Sustainable Energy and Mobility (STEMS), 10135 Turin, Italy
*   Correspondence: laura.bardi@crea.gov.it

**Abstract:** Kiwifruit has, for a long time, been widely affected by a syndrome named "kiwifruit decline syndrome" (KiDS). Several environmental factors have already been investigated looking for the possible origin of this syndrome. Recently, a possible role of climate change has been proposed, highlighting the influence of high air and soil temperature. In this work, the role of rising vapor pressure deficit (VPD) was also investigated in an experimental orchard in which several agronomic practices were examined in order to find strategies to face KiDS occurrence in crops. Stomatal control in response to rising VPD showed to be lacking, and root xylem vessel size and number modifications were observed as an adaptation to water stress; then, a scarce prospect of success facing sudden and strong weather events related to climate change can be expected in this plant. None of the agronomic strategies tested, that were focused on the soil quality improvement and on prevention of desiccation, avoided the KiDS occurrence. Agronomic management should move to new practices focused on orchard climate control.

**Keywords:** *Actinidia chinensis* var. *deliciosa*; kiwifruit decline syndrome; climate change; abiotic stress; VPD; leaf gas exchanges; leaf water potential; xylem anatomy; agronomic management; soil

## 1. Introduction

Several crops are showing symptoms of severe stresses induced by climate change, that is responsible for progressive modification of the local environment to which cultivated plants are adapted [1,2]. Global warming is inducing, exponentially with temperature, the rising of vapor pressure deficit (VPD) that causes great stresses on plants and can even give rise to plant mortality. Then, high temperatures and VPD are considered among the most important factors causing plant decline and reduced crops productivity following climate changes [3]. Plant adaptive response to climate change is variable among species, and also within the same species [4]. The more frequent adaptive behaviors are the reduced sensitivity of stomatal conductance to VPD, to prevent the arrest of photosynthesis [5], and the decrease in xylem vessels diameter, as a strategy to prevent the embolism caused by high transpiration following temperature rising [6]. This effect can be easily seen in perennial plants by observing the annual rings of stem sections. However, when high and rapid increase in temperature and high VPD induce a sudden increase in transpiration, low vessels diameter can hinder sap flow from roots to leaves; as a consequence, leaves temperature can rise even beyond the limit of desiccation.

Kiwifruit is a climbing, perennial, deciduous fruiting plant, originally growing in the wild in hills and mountains of southern and central China. During the XXth Century it was introduced and cultivated in New Zealand, then in United States and Europe, where it became an important crop. Nowadays, it is widely cultivated in China, but also in New Zealand, Italy, Iran, Greece and Chile, and it is present in France, Turkey, Portugal, Japan and Spain as well [7]. For longtime kiwifruit orchards were not subjected to significant disease; however, important adversities have recently appeared, in particular a bacterial disease induced by *Pseudomonas syringae* pv. *actinidiae* (PSA), isolated in Japan in 1984 and in Italy eight years later, becoming pandemic in 2008 [8]; and the insect *Halyomorpha halys* or brown marmorated stink bug, has diffused in United States and Europe since 1998 [9], severely compromising the productivity and fruit quality. Since 2012, a new syndrome, called kiwifruit decline syndrome (KiDS), appeared in North Italy and gradually has spread in other Italian regions, causing a dramatic damage to the whole fruit sector in the affected areas. Early symptoms are petiole wilt and leaf desiccation; severe damages affect the root system, that appears reduced in size, with a brown discoloration of the stele and thin, mushy cortical tissues [10–12]. The symptomatic plants usually do not recover but collapse and die during the same or the following year. Several attempts were made to find the causes, from biotic or abiotic origin, and the possible remedies, but with poor results. Since the onset of KiDS, the attention was pointed almost exclusively toward soil and plant roots, where the more severe damages were evident. Much work was conducted looking for soilborne pathogenic microorganisms or abiotic stress factors, including agronomic practices, but none of the investigated factors was the cause of KiDS when examined alone [11,13–15]. As kiwifruit roots are highly sensitive to hypoxic conditions [11,16], watering management practices aimed at avoiding flooding were generally adopted, but even this did not stop KiDS spreading [11,16–19]. Several attempts have been made to improve the soil quality and to create best conditions for roots wellness [10,11,18,20]; however, the best soil and water management practices were also ineffective in preventing the KiDS onset. Nowadays, KiDS is commonly considered a multifactor syndrome still unexplained.

Kiwifruit is a liana, characterized as long, flexible, with relatively small cross-sectional area stems, by very wide xylem vessels (up to 0.5 mm diameter), and by very large and numerous leaves [21–23]; this makes it particularly exposed to possible damages induced by drought. Moreover, its roots are highly sensitive to hypoxia induced by flooding, even if short lasting; as a consequence, it is not easy to meet the water needs of this plant without occurring in the risk of root damage due to low oxygen availability.

During the last decades, significant changes in climatic conditions were observed in Italy; notably, very high temperatures in summer, and long drought times interrupted by extreme meteorological events are more and more frequently happening. The hypothesis that climate change can contribute to the onset and the worsening of KiDS was recently suggested [15]. High temperature, in particular soil temperature, has been proposed as a possible cause of severe stress in roots, due to low availability of photoassimilates and consequent damage to cambial activity, with anatomical modifications and limited root growth and turnover as a consequence [10,15,18,24,25]. High temperature is also related to high VPD as it depends on the difference between saturation water vapor pressure (the maximum water vapor that air can hold at a certain temperature) and actual vapor pressure (the vapor really present in air at the same temperature); VPD is considered an indicator of the atmospheric desiccation strength [3]. Then, VPD could play a major role in kiwifruit, that is particularly sensitive to drought stress due to its botanical features. In spite of ascertained role of VPD, its influence in KiDS onset has not yet been explored. In this work, several agronomic managements were compared in an experimental kiwifruit orchard planted in the North Italy (Piedmont), in an area strongly affected by KiDS, looking for possible practices effective in preventing severe plant damages. The implemented measures, mainly focused on the soil quality improvement in order to promote the roots aeration and wellness and to stimulate plant response to stress, and on the possible guard against desiccation by the addition of glycine-betaine as an osmo-protectant, included:

- Soil ridging, to improve roots aeration and avoid waterlogging;
- Addition of compost to soil, to mainly improve soil physical properties;
- Addition of commercial microbial consortia to soil, to improve soil biological fertility and promote plant response to stress;
- Addition of zeolite, to improve water turnover in soil;
- Addition of glycine-betaine, to protect leaves from desiccation.

The growth, health and physiological behavior of the plants were detected throughout the growing season. The findings were also related to the monitored environmental parameters, with particular focus on VPD to investigate its possible role in KiDS onset.

## 2. Materials and Methods

### 2.1. Experimental Design and Trial Set Up

An experimental orchard was established in a flat, traditionally fruit-growing, strongly affected by KiDS area in Piedmont. New kiwifruit (cv Hayward) plantation (Figure S1) was carried out in 2017 in a field (Figure S2) previously hosting a 30-years old kiwifruit orchard, uprooted following a severe KiDS occurrence (Figure S3), in Saluzzo (CN, 44°38′49.8″ N 7°31′52.6″ E). The soil was loamy [26], neutral (pH 6.8), with 2.9% organic matter, 87 ppm exchangeable K, 64 ppm soluble P and 2409 ppm exchangeable Ca content. Plants from in vitro cultures of meristems were employed in order to avoid the risk of spreading pathogens.

The trial was set up as a block design. The orchard was divided in four blocks, and designed perpendicularly into rows. Each row corresponded to a treatment; plants were randomly selected inside each block in order to have four replicates per 11 different treatments:

FlCon: flat soil, without any addition;

FlZeo: flat soil, with addition of zeolite;

FlCom: flat soil, with addition of compost (130 t ha$^{-1}$);

FlGly: flat soil, with addition of glycine-betaine;

TiCon: soil tilled to form raised ridges of about 50 cm in rows, without any addition;

TiCom: soil tilled to form raised ridges of about 50 cm in rows, with the addition of compost (80 t ha$^{-1}$);

TiMi1: soil tilled to form raised ridges of about 50 cm in rows, with the addition of microbial consortium 1;

TiMi2: soil tilled to form raised ridges of about 50 cm in rows, with the addition of a microbial consortium 2;

TiMi1Gly: soil tilled to form raised ridges of about 50 cm in rows, with the addition of a microbial consortium 1 and addition of glycine-betaine;

TiMi2Gly: soil tilled to form raised ridges of about 50 cm in rows, with the addition of a microbial consortium 2 and addition of glycine-betaine;

TiZeo: soil tilled to form raised ridges of about 50 cm in rows, with the addition of zeolite.

The orchard was managed for three years according to usual agronomical practices. Localized irrigation was carried out by a drip irrigation system with watering volumes calculated on the basis of monitored environmental parameters to maintain the soil water potential near the field capacity (33 cbar).

### 2.2. Materials

#### 2.2.1. Compost

The compost (ACEA Pinerolese Industriale S.p.a., Pinerolo (TO), Italy), containing 23% organic C, 2.55% N (93% in organic forms), 1.6% $P_2O_5$ and 1.2% $K_2O$, pH 8.2, was added before planting.

### 2.2.2. Microbial Consortium 1

Commercial microbial consortium Micosat (CCS Aosta, Quart (AO), Italy), composed of 40% mycorrhizal fungi, 21.6 % saprophytic fungi and $4.85 \times 10^7$ C.F.U. $g^{-1}$ plant growth promoting bacteria, was added at a dose of 10 g·plant$^{-1}$. 40 mL of a mixture composed by 20 mL of Eutrofit (AGM s.r.l., Castelnovo Di Sotto (RE), Italy), an organic N fertilizer (4% N, 14% C) and 20 mL Red radicale (Germina s.r.l., Montepulciano (SI), Italy), a humic extract (0.5% N, 1.5% C), were added to each plant, as a supplement supporting the establishment of the microbial inoculum. The treatment was repeated three times each year: at planting, after one month, and after two months from planting for a total of nine treatments.

### 2.2.3. Microbial Consortium 2

Commercial microbial consortium Ekoprop (Green Ravenna s.r.l., Ravenna (RA), Italy), composed of 1% mycorrhizal fungi, $5 \times 10^5$ C.F.U. $g^{-1}$ saprophytic fungi and $1.6 \times 10^6$ C.F.U. $g^{-1}$ plant growth promoting bacteria, was added at a dose of 3 g plant$^{-1}$ simultaneously with 15 mL plant$^{-1}$ rooting inducer Rootmost (10% algae extract, 1% P, 2.5 % K, vitamins B1, B6, B12, microelements; Green Ravenna s.r.l., Ravenna (RA), Italy). The treatment was repeated three times each year (mid-April, mid-May, mid-June).

### 2.2.4. Zeolite

A total of 500 g of Chabasite (Bal-co S.p.a., Sassuolo (MO), Italy) for each plant were added and mixed to soil at the time of planting.

### 2.2.5. Glycine-Betaine

A commercial preparation, Betamax (Green Ravenna s.r.l., Ravenna (RA), Italy), containing 99% glycine-betaine, was administered at the dose of 2 g plant$^{-1}$ three times each year during growing season (mid-April, mid-May, mid-June).

### 2.3. Evaluated Parameters

### 2.3.1. Leaf Gas Exchanges

Photosynthesis (A, mmol $CO_2$ m$^{-2}$ s$^{-1}$), stomatal conductance (gs, mol $H_2O$ m$^{-2}$ s$^{-1}$), transpiration (E, mmol $H_2O$ m$^{-2}$ s$^{-1}$), and substomatal $CO_2$ concentration (ci, mmol $CO_2$ mol$^{-1}$) were measured by InfraRed Gas Analyser (IRGA, Portable Photosynthesis system Li-Cor, Ecosearch, Montone (PG), Italy). Water use efficiency (WUE, mmol $CO_2$ mmol $H_2O^{-1}$) was also calculated as A/E.

The measures were carried out during the second year from plantation, in mid-June (20 days after mid bloom, DAMB), mid-July (50 DAMB) and mid-August (80 DAMB), in the morning between 8:30 and 11:00 a.m. The measurements were carried out on four plants per treatment, and for each plant on two leaves: one apical (young, fully developed) leaf, and one basal leaf. Based on the obtained results, the measures were repeated during the third year in July (50 DAMB), as this was chosen as the most significant time to evidence differences among the treatments and to observe the physiological behaviour of plants under heat and water stress.

### 2.3.2. Leaf Water Potential

Midday Leaf Water Potential (MPa, MLWP,) was measured in the second year from plantation. The measurements were carried out in mid-June (20 DAMB) and mid-July (50 DAMB), in four plants per treatment, from 12:00 to 15:00, and for each plant in two leaves: one apical leaf (young, fully developed leaf), and one basal leaf, with a pressure chamber Scholander (Ecosearch s.r.l., Montone (PG), Italy) according to [27,28].

### 2.3.3. Stem Growth

Stem growth was evaluated in all plants (about 300) in the second and third year as the % increase in the collar diameter, that was measured with a calibre at the start (−40 DAMB), and at the end (120 DAMB) of the vegetative season.

### 2.3.4. PSA and KIDS Symptoms

The occurrence of the bacterial disease PSA and KiDS was quantified by checking the number of affected plants among all the plants of the orchard (about 300) and reported as percentage of affected plants for each treatment. The bacterial disease PSA induces symptoms that are easily distinguishable from KiDS symptoms: the bacterial disease PSA induces cortical necrosis, exudates and leaf spots, that appear very early at the vegetative restart in springtime (Figure S4), while KiDS symptoms (petiole wilt, leaves chlorosis and desiccation, Figure S3) appear later in summer [8,10]. Bacterial disease PSA symptoms were checked at −40 DAMB and KiDS symptoms were checked at 120 DAMB.

Randomly selected samples of soils and roots were collected and analysed at the start and at the end of the experimental period, allowing us to exclude the presence of soilborne pathogens (unshown data).

### 2.3.5. Microscopic Analysis of Roots Sections

At the end of vegetative season of third year, samples of roots were taken from plants of each treatment. Roots of about 1 cm diameter were cut in segments of about 3 mm thickness with a steel razor blade. These samples were observed by stereoscopic microscopy (Optika Microscopes s.r.l., Ponteranica (BG), Italy) and by Scanning Electron Microscopy.

For light microscopy, small root cubes of approximately 1–2 $cm^3$ were obtained. Transverse sections of 20 μm thickness were cut using a rotary microtome (Leica). After bleaching and rinsing with water, the sections were stained with Lugol's iodine solution (5% iodine, 10% potassium iodide, bi-demineralized water) to stain starch grains that indicate where carbohydrates are stored within the tissue [29].

Sections were acquired using a light microscope DM 2000 LED (Leica, Wetzlar, Germany), photographed with a digital camera (Leica DFC450 C), observed, and analysed with Leica LAS X software.

For Scanning Electron Microscopy (SEM), root samples were placed on an aluminium stub and coated with few nanometers of gold (about 10–15 nm) in order to prevent any charging effect during the analysis. A Zeiss EVO 50 XVP scanning electron microscope equipped with $LaB_6$ source was used for the analysis. The microscope is endowed with detectors for secondary and backscattered electrons for image acquisition and Energy Dispersion Spectroscopy (EDS) for elemental analysis.

### 2.3.6. Weather Data and Soil Water Potential

Weather data were recorded throughout the three years (2017–2019) by a METOS® station (Pessl Instruments GmbH, Weiz, Austria) located into the experimental orchard, that directly provided VPD data.

Soil water potential was measured using tensiometers (Watermark, Irrometer Company Inc., Riverside, CA, USA) positioned at 10 cm, 20 cm, 30 cm, 40 cm depth in the FlCon and TiCon treatments; these data were daily used to calculate the watering volumes in order to maintain the optimal plant water supply, avoiding the risk of roots flooding.

### 2.3.7. Processing of Data and Statistical Analysis

Data were processed by Excel (MS Software) and PAST 4.03 Software. ANOVA and least significant difference (LSD) analyses were carried out by assuming a *p*-value threshold ≤0.05.

## 3. Results

### 3.1. Stem Growth

The stem growth, measured as a % of increase in the stem collar diameter during second- and third-year growing season, is reported in Figure 1. In general, a decrease in growth rate was observed from second to third year. High growth rate induced by composting during the second year was drastically reduced in both flat and ridged soil during third year. Flat soil strongly prevented growth during third year, when the growth

was higher in treatment TiCon. The growth rate decline from second to third year was prevented by the microbial consortium 1 (TiMi1), that also showed the best-developed and fibrous roots-rich root system in plants uprooted at the end of third year (Figure S5). A positive role of glycine-betaine was also observed (Figure 1 and Figure S5), in particular when TiMi2 is compared to TiMi2Gly treatment (Figure 1): in these treatments, roots systems did not appeared significantly different (Figure S5), so it can be postulated that microbial consortium 1 has a direct role to play in roots wellness, as well as emphasizing the role of glycine-betaine on the leaves protection from water stress.

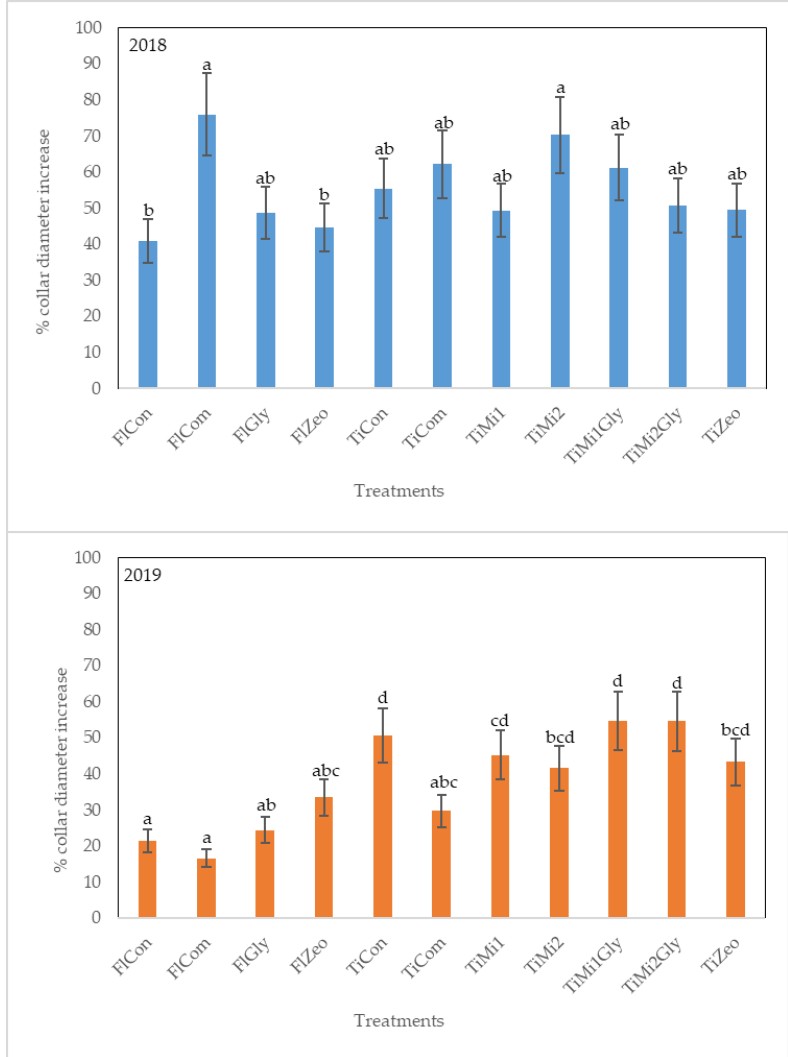

**Figure 1.** Stem growth expressed as % collar diameter increase during growing season. Data are the mean of 23 replicates. The letters on the columns show significant differences between the treatments according to ANOVA, least significant difference (LSD) analyses at $p$-value threshold $\leq$0.05. FlCon: flat soil, without any addition; FlCom: flat soil, with addition of compost (130 t ha$^{-1}$); FlGly: flat soil, with addition of glycine-betaine; FlZeo: flat soil, with addition of zeolite; TiCon: soil tilled to form raised ridges of about 50 cm in rows, without any addition; TiCom: soil tilled to form raised ridges of about 50 cm in rows, with addition of compost (80 t ha$^{-1}$); TiMi1: soil tilled to form raised ridges of about 50 cm in rows, with addition of microbial consortium 1; TiMi2: soil tilled to form raised ridges of about 50 cm in rows, with addition of a microbial consortium 2; TiMi1Gly: soil tilled to form raised ridges of about 50 cm in rows, with addition of a microbial consortium 1 and addition of glycine-betaine; TiMi2Gly: soil tilled to form raised ridges of about 50 cm in rows, with addition of a microbial consortium 2 and addition of glycine-betaine; TiZeo: soil tilled to form raised ridges of about 50 cm in rows, with addition of zeolite.

### 3.2. Bacterial Disease PSA and KiDS Symptoms

The bacterial disease PSA and KiDS symptoms detected during third year growing season are reported in Table 1 as percentage of affected plants for each treatment.

**Table 1.** Bacterial disease PSA and KiDS symptoms occurrence expressed as affected plants percentage of 23 total plants detected per each treatment at the end of third year vegetative season.

| Treatment | Bacterial Disease PSA Symptoms (% Affected Plants) | KiDS Symptoms (% Affected Plants) |
|---|---|---|
| FlCon | 16.7 | 100 |
| FlCom | 42.1 | 100 |
| FlGly | 20.0 | 100 |
| FlZeo | 9.5 | 100 |
| TiCon | 15.0 | 91 |
| TiCom | 55.9 | 66.7 |
| TiMi1 | 20.0 | 91.7 |
| TiMi2 | 18.4 | 100 |
| TiMi1Gly | 16.7 | 90.9 |
| TiMi2Gly | 18.2 | 90 |
| TiZeo | 19.0 | 91.7 |

It can be observed that at the end of the third year KiDS symptoms were present in all treatments. In flat soil treatments, all plants were affected by KiDS, and several dead plants were detected, while the incidence of KiDS was slightly lower in ridged treatments, where no dead plants were found. Composting increased the occurrence of PSA.

### 3.3. Leaf Water Potential

No significant differences were detected among treatments and among young/old leaves for MLWP neither in June nor in July. In general, very low values were detected, that can be considered a possible indication of a state of severe water stress [30], in spite of watering volumes carefully measured in order to supply the water needed without excess. The highest value (−0.75 MPa) was detected in June (30 DAMB) in apical leaves of treatment FlZeo. In July (50 DAMB), a further decrease was observed (unshown data), with the stronger decrease (60%) detected in TiCon treatment. The mean data detected for all (basal and apical) leaves and treatments in June are reported in Figure 2, showing an early onset of water stress status.

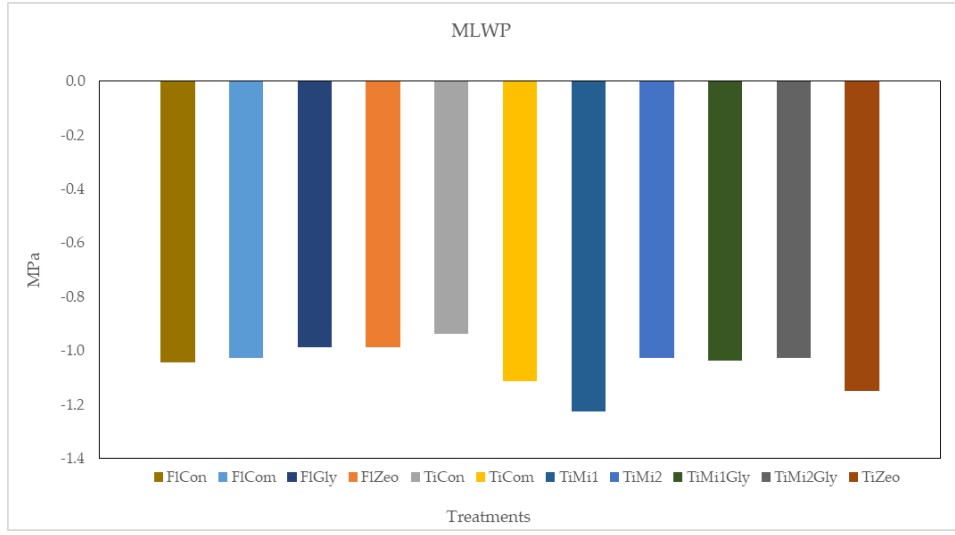

**Figure 2.** Midday Leaf Water Potential (June, 20 DAMB, second year). Data are the mean of 4 replicates. No significant differences were detected between the treatments by ANOVA. FlCon: flat

soil, without any addition; FlCom: flat soil, with addition of compost (130 t ha$^{-1}$); FlGly: flat soil, with addition of glycine-betaine; FlZeo: flat soil, with addition of zeolite; TiCon: soil tilled to form raised ridges of about 50 cm in rows, without any addition; TiCom: soil tilled to form raised ridges of about 50 cm in rows, with addition of compost (80 t ha$^{-1}$); TiMi1: soil tilled to form raised ridges of about 50 cm in rows, with addition of microbial consortium 1; TiMi2: soil tilled to form raised ridges of about 50 cm in rows, with addition of a microbial consortium 2; TiMi1Gly: soil tilled to form raised ridges of about 50 cm in rows, with addition of a microbial consortium 1 and addition of glycine-betaine; TiMi2Gly: soil tilled to form raised ridges of about 50 cm in rows, with addition of a microbial consortium 2 and addition of glycine-betaine; TiZeo: soil tilled to form raised ridges of about 50 cm in rows, with addition of zeolite.

### 3.4. Leaf Gas Exchanges

No significant differences were detected among treatments for leaf gas exchanges when data from apical and basal leaves all together or from basal leaves alone were analyzed. Significant differences were observed for apical leaves when analyzed alone, as shown in Figure 3, where the apical leaf gas exchanges measured in July, in the mid of the vegetative season (50 DAMB) of the third year, are reported: the highest values of gs, A and E were observed in treatment TiMi2, and the lowest in FlZeo. Substomatal $CO_2$ concentration was lowest in TiMi1Gly and TiZeo, highest in TiMi1.

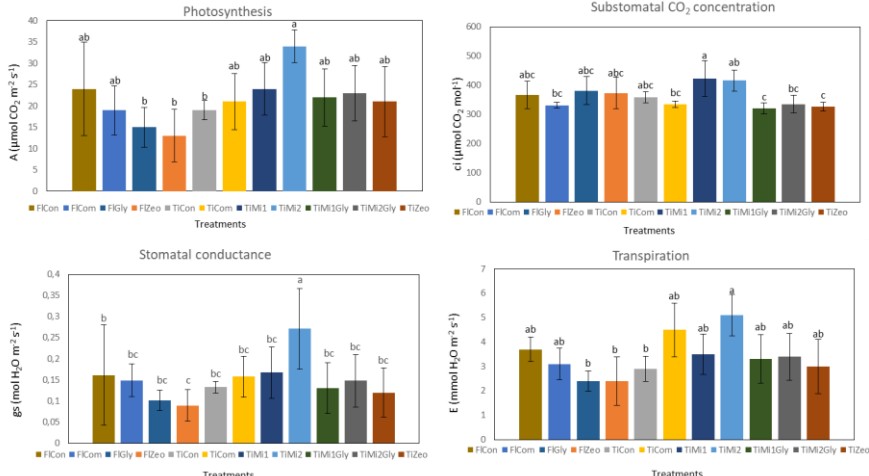

**Figure 3.** Apical leaf gas exchanges detected on July, 50 DAMB, third year, in the morning between 8:30 and 11:00 a.m. Data are the mean of 4 replicates. The letters on the columns show significant differences between the treatments according to ANOVA, least significant difference (LSD) analyses at *p*-value threshold ≤0.05. Error bars represent standard deviation. FlCon: flat soil, without any addition; FlCom: flat soil, with addition of compost (130 t ha$^{-1}$); FlGly: flat soil, with addition of glycine-betaine; FlZeo: flat soil, with addition of zeolite; TiCon: soil tilled to form raised ridges of about 50 cm in rows, without any addition; TiCom: soil tilled to form raised ridges of about 50 cm in rows, with addition of compost (80 t ha$^{-1}$); TiMi1: soil tilled to form raised ridges of about 50 cm in rows, with addition of microbial consortium 1; TiMi2: soil tilled to form raised ridges of about 50 cm in rows, with addition of a microbial consortium 2; TiMi1Gly: soil tilled to form raised ridges of about 50 cm in rows, with addition of a microbial consortium 1 and addition of glycine-betaine; TiMi2Gly: soil tilled to form raised ridges of about 50 cm in rows, with addition of a microbial consortium 2 and addition of glycine-betaine; TiZeo: soil tilled to form raised ridges of about 50 cm in rows, with addition of zeolite.

Water use efficiency was also calculated as an index of water stress. In all treatments, the detected values showed a severe water stress status (Figure 4): in an assay recently carried out in South China, the highest WUE detected in kiwifruit under water deficit was

1.32 [31]. Furthermore, WUE strongly rose from second to third year in all the treatments: this can be considered a symptom of a progressive water stress status of plants, even if suitable water volumes were always delivered by irrigation to maintain the soil water potential near the field capacity.

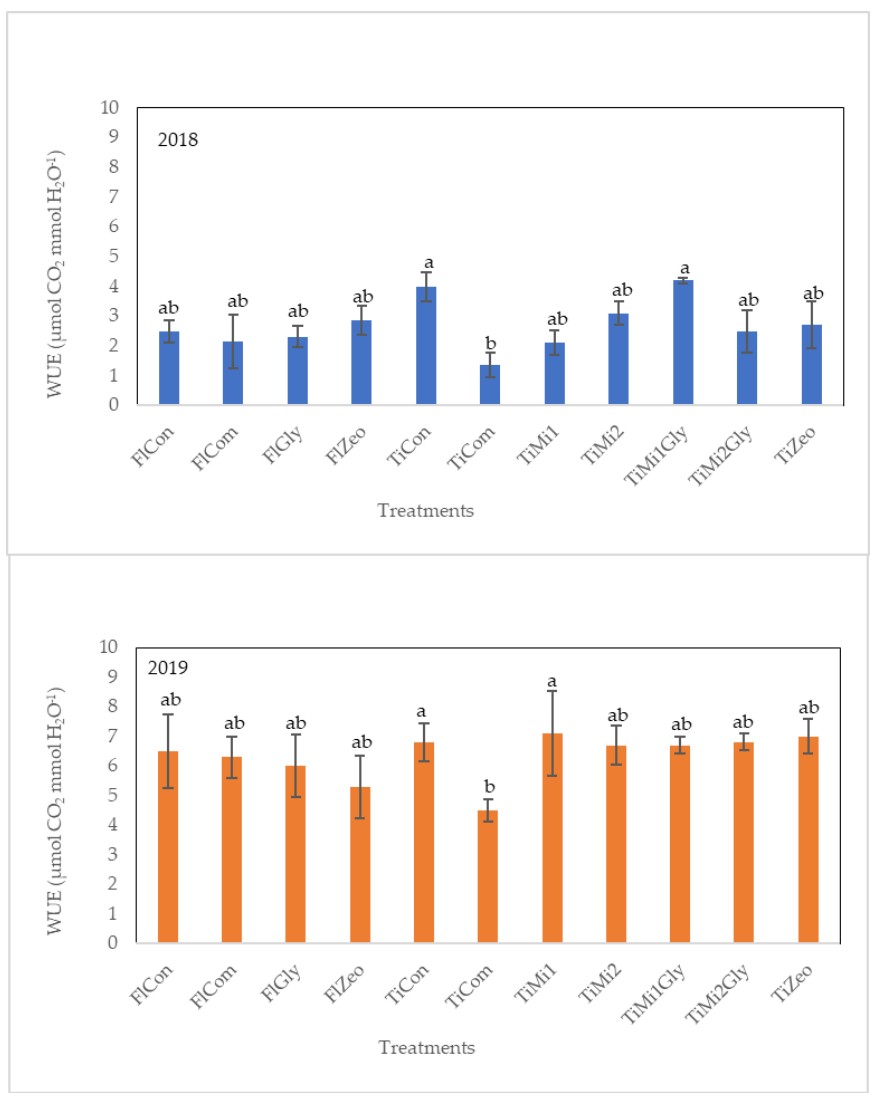

**Figure 4.** Water Use Efficiency (WUE, calculated as A/E checked in the morning between 8:30 and 11:00 a.m. in July, 50 DAMB, third year). Data are the mean of 4 replicates. The letters on the columns show significant differences between the treatments according to ANOVA, least significant difference (LSD) analyses at *p*-value threshold ≤0.05. Error bars represent standard deviation. FlCon: flat soil, without any addition; FlCom: flat soil, with addition of compost (130 t ha$^{-1}$); FlGly: flat soil, with addition of glycine-betaine; FlZeo: flat soil, with addition of zeolite; TiCon: soil tilled to form raised ridges of about 50 cm in rows, without any addition; TiCom: soil tilled to form raised ridges of about 50 cm in rows, with addition of compost (80 t ha$^{-1}$); TiMi1: soil tilled to form raised ridges of about 50 cm in rows, with addition of microbial consortium 1; TiMi2: soil tilled to form raised ridges of about 50 cm in rows, with addition of a microbial consortium 2; TiMi1Gly: soil tilled to form raised ridges of about 50 cm in rows, with addition of a microbial consortium 1 and addition of glycine-betaine; TiMi2Gly: soil tilled to form raised ridges of about 50 cm in rows, with addition of a microbial consortium 2 and addition of glycine-betaine; TiZeo: soil tilled to form raised ridges of about 50 cm in rows, with addition of zeolite.

Stomatal conductance (gs, mol $H_2O$ m$^{-2}$ s$^{-1}$) showed a great variability throughout the growing season, in different treatments and in different years. All the stomatal conductance data measured throughout the second and third years (500 data recorded) were used to check the relation between gs and VPD, as reported in Figure 5. While in most plant species stomatal conductance decreases with increasing VPD as a defense strategy against desiccation, it is immediately evident that all kiwifruit plants in the experimental orchards showed a very low stomatal sensitivity to VPD, independently on the treatment: this means that kiwifruit is strongly exposed to severe danger of cavitation and leaves desiccation under water stress and that it shows an anisohydric behavior [5].

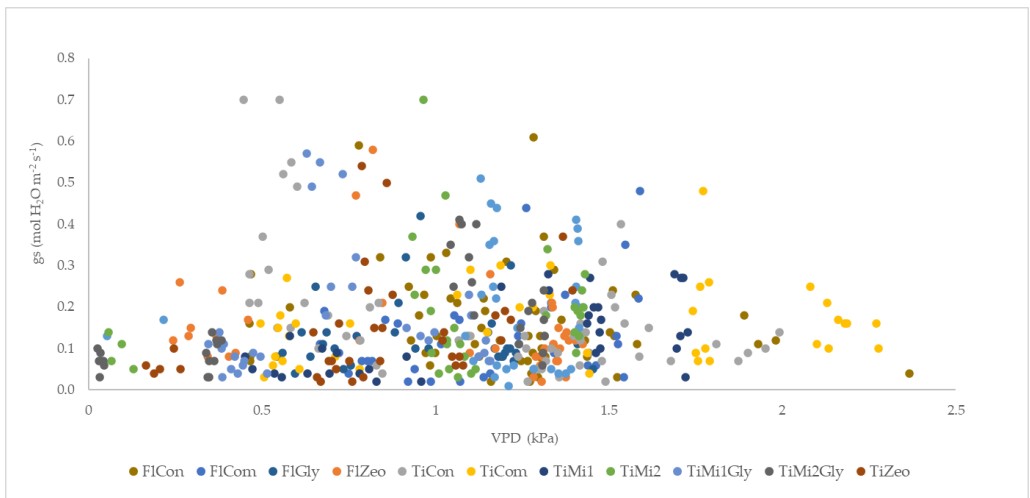

**Figure 5.** Stomatal conductance (gs, mol $H_2O$ m$^{-2}$ s$^{-1}$) response to increasing vapor pressure deficit, VPD (kPa). Data recorded during the second and third year, in the morning between 8:30 and 11:00 a.m., between 20 and 80 DAMB. FlCon: flat soil, without any addition; FlCom: flat soil, with addition of compost (130 t ha$^{-1}$); FlGly: flat soil, with addition of glycine-betaine; FlZeo: flat soil, with addition of zeolite; TiCon: soil tilled to form raised ridges of about 50 cm in rows, without any addition; TiCom: soil tilled to form raised ridges of about 50 cm in rows, with addition of compost (80 t ha$^{-1}$); TiMi1: soil tilled to form raised ridges of about 50 cm in rows, with addition of microbial consortium 1; TiMi2: soil tilled to form raised ridges of about 50 cm in rows, with addition of a microbial consortium 2; TiMi1Gly: soil tilled to form raised ridges of about 50 cm in rows, with addition of a microbial consortium 1 and addition of glycine-betaine; TiMi2Gly: soil tilled to form raised ridges of about 50 cm in rows, with addition of a microbial consortium 2 and addition of glycine-betaine; TiZeo: soil tilled to form raised ridges of about 50 cm in rows, with addition of zeolite.

*3.5. Microscopic Analysis of Roots Sections*

Stereoscopic microscopy observation evidenced significant differences among roots sampled from plants subjected to different treatments; one sample collected from a private garden, from a perfectly healthy kiwifruit plant (Hayward), was also analyzed in order to obtain a certainly healthy root for comparison. Some of the already well-known symptoms of KiDS, such as detachment of cortex from the central stele and decay of phloem and cambial layers in roots, were present in almost all samples, but more evident in some treatment, such as FlCon (Figure 6b). Interesting differences were observed in the number and diameter of xylem vessels. In the healthy root (Figure 6a), very wide xylem vessels were evident; on the contrary, increased number, corresponding to decreased diameter, were observed in roots from treatment in which KiDS symptoms were present, but not severe (Treatments TiCon and TiCom, Figure 6c,d), while a decreased number associated to decreased diameter was observed in treatments with severe KiDS symptoms (Treatment FlCon, Figure 6b).

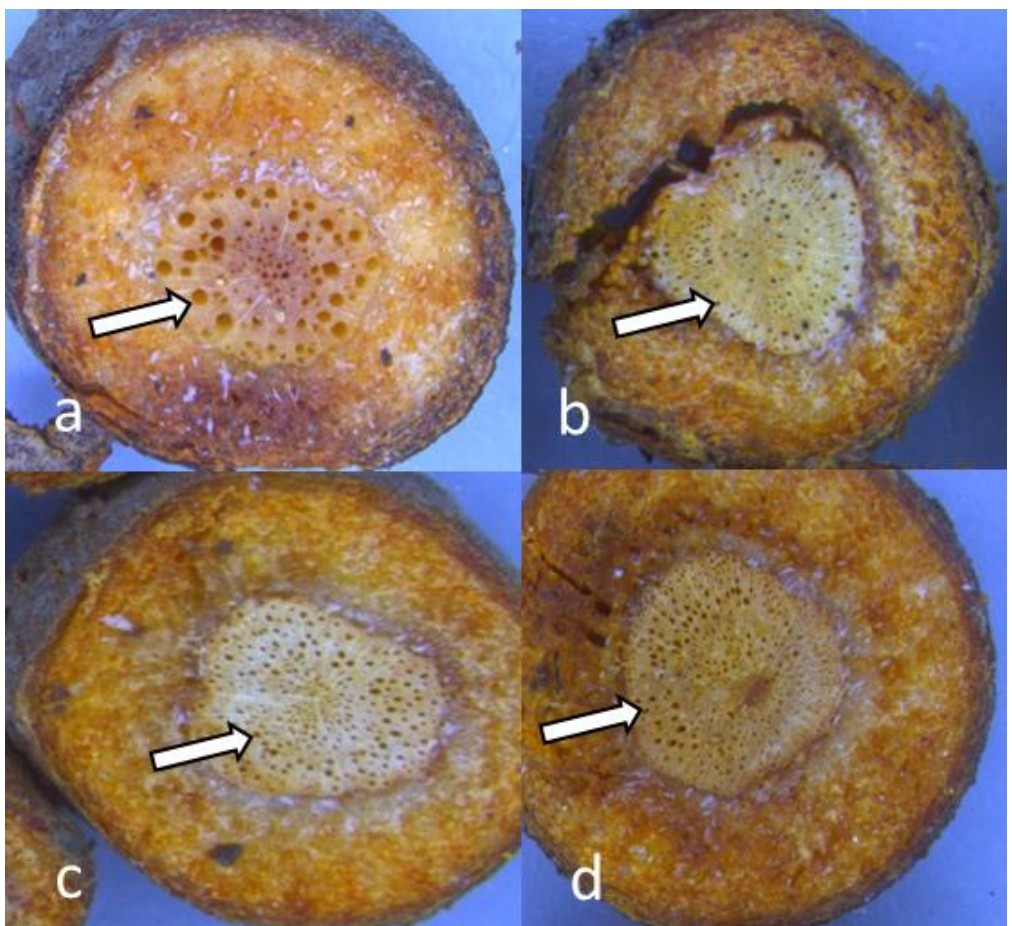

**Figure 6.** Roots section observed in stereoscopic microscopy. (**a**) = root from a perfectly healthy kiwifruit plant; (**b**) = root from treatment FlCon; (**c**) = root from treatment TiCon; (**d**) = root from treatment TiCom. Arrows point xylem vessels. Differences between treatments are mainly in the number and diameter of xylem vessels.

Plants showing a decreased number and diameter of xylem vessels also showed a small-developed root system with few fibrous roots when they were uprooted at the end of the third year (Figure S5).

Light microscopic analysis of roots sections stained with Lugol solution showed the presence of a fair amount of starch mainly in radial rays of xylem parenchyma in most of treatments, with some slight difference among them. As shown in Figure 7, parenchymal cells surrounding xylem vessels also contain starch, but they appear partially depleted. This could be considered a symptoms of water stress, with starch used to protect vessels from cavitation and to empower sap flow [6].

SEM analysis of roots sections allowed us to highlight the presence of occlusions in xylem vessels in plants of treatment FlCon, that were not observed in other samples (Figure 8).

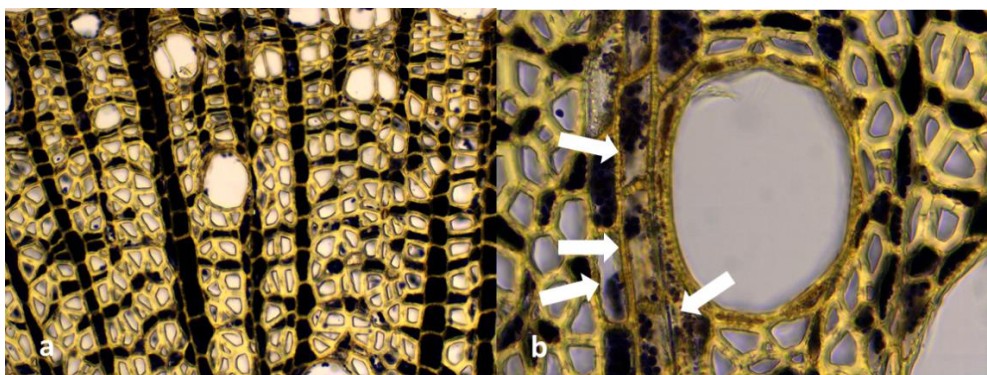

**Figure 7.** Sections of roots sampled from plants of treatment TiMi1, stained with Lugol's solution and observed in light microscopy. Starch is evidenced in dark in parenchymatous cells rays (**a**). Arrows indicate partial starch depletion in cells surrounding xylem vessels (**b**).

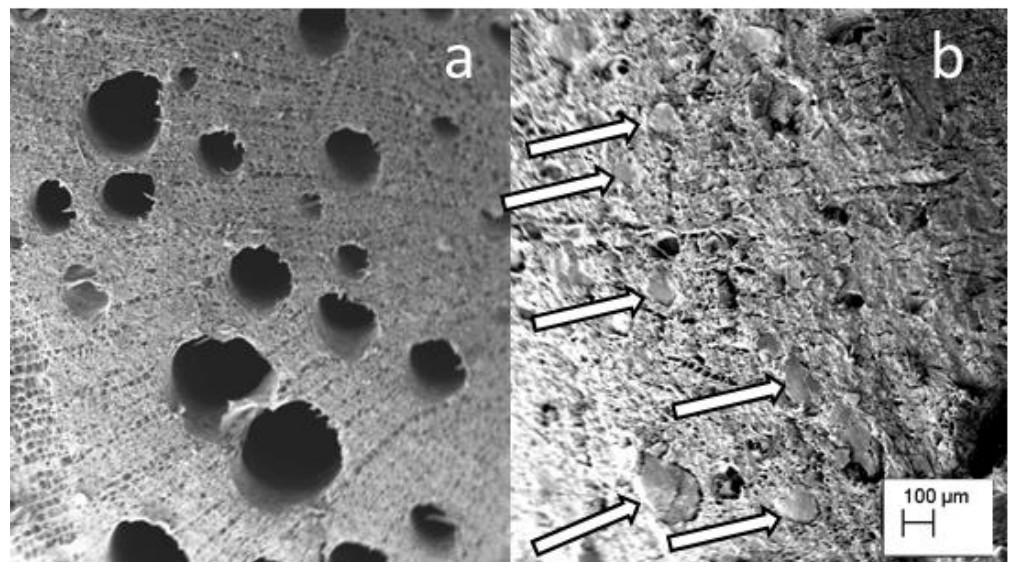

**Figure 8.** Roots sections observed by scanning electron microscope (SEM); (**a**)—root from a healthy plant (private garden); (**b**)—root from treatment FlCon. Arrows indicate some of the occluded vessels.

*3.6. VPD*

During the three years of the trial, the VPD reached high values during growing season. In 2017 high VPD values were recorded already on 24 may (3.32 kPa) and again on 10 and 16 June, in 13–26 July and 4–6 August (until 3.50 kPa). In 2018, the highest values were recorded on 30 June (3.30–3.56 kPa), on 17–23 July and on 22 August (3.00–3.08 kPa). In 2019 it reached values between 3.45 and 4.66 kPa in late afternoon at the end of June, and again during first days of July (3.09–3.25 kPa). The number of days with VPD higher than 3 were 16, 13 and 16 in 2017, 2018 and 2019, respectively. In the morning, when leaf gas exchanges were measured, the VPD detected was between 0.03 and 2.37 (Figure 5). The daily maximum VPD values recorded from May 2017 to December 2019 are reported in Figure 9.

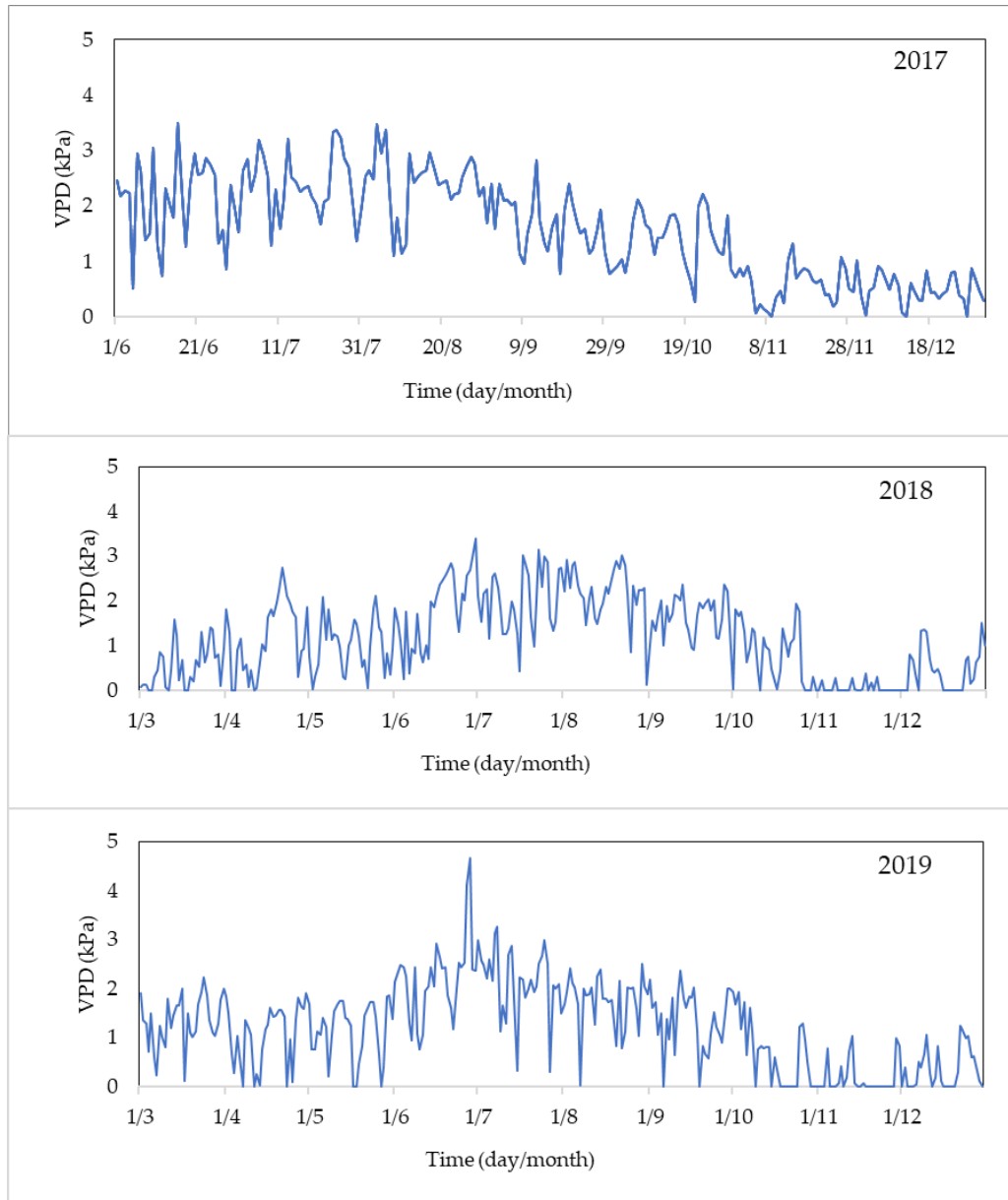

**Figure 9.** Vapor pressure deficit VPD (kPa) daily maximum values recorded throughout the three years (2017–2019) by a METOS® station (Pessl Instruments GmbH, Weiz, Austria) located into the experimental orchard.

## 4. Discussion

Global warming is inducing an exponential rise of VPD, and this causes great stresses on plants due to increased transpiration rate and water loss from soil. VPD can even give rise to plant mortality independently on other factors associated with climate change, such as high temperature or drought. Indeed, VPD is nowadays considered one main abiotic stress factor for plants due to climate change [3].

A possible mechanism of adaptation to rising VPD is the modification of xylem anatomy, with a decrease in the xylem vessel diameter, to reduce vulnerability to cavitation [6,32–34]. Kiwifruit is characterized by very wide xylem vessels and by numerous and large leaves; this makes it very sensitive to high evaporative demand caused by high temperature, VPD and drought stress [21]. In our trial, we observed a reduced diameter in almost all treatments when compared to a healthy plant (Figure 6). Reduced xylem vessels diameter is a good means of adaptation to a long-term climatic change, such as rising

VPD; however, it can cause a reduced sap flow rate, that can become insufficient when sudden and severe water stress or rapidly increasing evaporative demand occur [25,32]. So, this strategy of adaptation to climate change can be considered the origin of water stress detected by MLWP and WUE measurements (Figures 2 and 4). In some cases, the reduced xylem vessels diameter was also associated with a reduced vessels number (i.e., plants from treatment FlCon); this could be due to the associated stresses induced by high VPD, high temperature and soil hypoxic conditions, that prevent cambial activity and xylogenesis [10]. Therefore, plants from treatment FlCon were particularly vulnerable; indeed, they showed the lowest growth (Figure 1) and the highest KiDS symptoms occurrence (Table 1).

The VPD monitored during the three years of the trial (Figure 9) reached very high values during growing season, with more than 13 days/year with VPD > 3 kPa. It showed a critical point in 2019, when 3 days with VPD > 4.0 kPa and 2 days with VPD > 4.5 kPa were detected, with the highest value reaching 4.66 kPa at 30 DAMB . High values were recorded in the same period also in 2018 (3.30–3.56 kPa) and in 2017, when high VPD values were recorded already during blooming (3.32 kPa). At the end of the trial, KiDS symptoms were present all over the orchard. MLWP revealed that the plants in all treatments, in spite of the careful watering management, were already under water stress during the growing season of the second year, starting early in June (Figure 2): indeed, Calderon-Orellana et al. [30] found MLWP values between −0.8 and −0.6 MPa in 2018 and between −1.0 and −0.6 Mpa in 2019 throughout the growing season under wet conditions, and the lowest MLWP value detected under regulated deficit irrigation was −1.46 Mpa, while in our trial, the highest detected value was −0.75 Mpa, and the lowest value reached −1.8 Mpa. Moreover, WUE detected at mid growing season (50 DAMB) indicated a general status of water stress in almost all treatments, rising over time, as shown in Figure 4. Plant species adapted to face drought stress and living in arid lands show rich starch deposition in roots parenchymal radial rays, as we also observed in our trial (Figure 7a). Partial depletion of starch from parenchymal cells surrounding xylem vessels (Figure 7b) could also be considered a symptom of water stress, as starch hydrolysis and sugar delivering to sap is a way to refill embolized xylem vessels by increasing its osmotic potential, then empowering sap flow to leaves [6,35–37].

Plant response to water stress induced by rising VPD can vary depending on species. In general, the most common adaptation of plants to water stress is stomatal closure, in order to reduce water loss due to transpiration and to prevent the risk of embolism in xylem vessels. However, as reduced photosynthesis is a consequence of stomatal closure, plants can develop long-term strategies for acclimation to rising VPD by reducing stomatal sensitivity to evaporative demand, in order to maintain active photosynthesis during heat waves [5,38]. Grossiord et al. [5] suggested that anisohydric plants will be more vulnerable to global warming than isohydric plants. Indeed, their reduced stomatal sensitivity to rising VPD will be counteracted by the negative carbon balance induced by high temperature; this will prevent the roots growth in depth, favoring under-developed and shallow root systems, that will not be able to respond to changes in soil moisture. For this reason, anisohydric plants will not even benefit correct soil watering management [3]. In our trial, kiwifruit exhibited these characteristics of high sensitivity to water stress: stomatal regulation was low, as indicated by gs/VPD relation (Figure 5), as typically happens in anisohydric plants. Moreover, plants of our trial showed shallow and under-developed root systems (Figure S5), as well as usually happens in plants from orchards affected by KiDS (Figure S3) [10,11,14,17,18,39]. The decline in plant growth rate over time (Figure 1) can be considered a consequence of the negative carbon balance too.

In addition to climate stress, kiwifruit is also endemically exposed to the biotic stress caused by the bacterial disease PSA, the only pathogenic microorganism detected also in our trial (Table 1). PSA is a bacterial disease which causes leaf spots and shoots canker, from which bacterial exudates leach (Figure S4) [8]. The probable systemic spreading of the bacterium in shoots via the xylem vessels has been proposed [40] by a mechanism already described, for example, for *Pseudomonas syringe* in *Prunus salicina* [41]. Other bacterial

pathogens have been described for their ability to use xylem vessels as paths to spread into the plant. Recently, the combination of biotic and abiotic stresses in an olive tree caused by climate change and *Xylella fastidiosa*, a bacterial disease causing the plant death, was studied [42–44]. In infected olive trees, transformed xylem anatomy was observed, with an increase in the number of vessels and in ring width, probably to enhance water conductivity damaged by xylem vessels occlusion, that is a consequence of both bacterial growth and plant response to infection [44]. Moreover, cultivar Leccino, resistant to *X. fastidiosa*, is poorly sensitive to cavitation due to xylem vessels low diameter, and it is characterized by efficient refilling mechanisms in response to cavitation, also due to a high storage of starch in xylem. On the other hand, sensitive cultivar Cellina di Nardò is characterized by the prevalence of wide xylem vessels [43]. Petit et al. [45] pushed forward the intriguing hypothesis that *X. fastidiosa* could worsen the cavitation risk in xylem by degrading pit membranes, in order to favor its own spreading from a xylem vessel to another, and also that embolized vessels could be a more favorable place for aerobic or semi-anaerobic bacteria than fully sap filled vessel.

This scenario, proved and proposed for *Xylella*—olive tree interaction under climate stress, could be probably extended to bacterial disease PSA—kiwifruit interaction, where kiwifruit shows anatomical and physiological similarities to *Xylella*-sensitive cultivar Cellina di Nardò. The presence of occlusions in xylem vessels was observed in our trial in root sections of plants from treatment FlCon (Figure 8b); these occlusions could be produced by the plant to constraint the bacterial pathogen spreading, as well as to isolate embolized vessels [46]. Then, biotic and abiotic stresses could add up, until plant death: this could also explain co-presence of KiDS-affected and healthy plants, apparently facing the same climatic stresses, as well as the survival or death of bacterial disease PSA-affected plants in different climatic areas.

Then, due to its physiological and anatomical features, kiwifruit can be highly sensitive to severe damages induced by climate change, showing related symptoms earlier than other fruit crops. Watering and agronomic managements that are commonly considered proper for fruit trees could be not appropriate for kiwifruit under climate stress. The treatments implemented in our trials, chosen among those considered the best in ensuring plant wellness and optimal environmental conditions, were not effective to prevent KiDS occurrence; however, some difference was observed in growth and physiological behavior of plants from different treatments, giving the perspective that some agronomic practices can be helpful to improve plant resilience to climate stress. Soil ridging is confirmed as almost essential for kiwifruit to ensure aeration necessary to roots system. The composting effect on stem growth was detrimental over time (Figure 1); the early growth induced at second year was probably related to the early high occurrence of bacterial disease PSA (Table 1). On the other hand, gas exchange parameters (Figure 3) and WUE (Figure 4) in treatment TiCom indicate a state of stress and occurrence of KiDS symptoms lower than in other treatments (Table 1 and Figure S5). Zeolite showed no benefits (treatments FlZeo and TiZeo). Glycine-betaine, used for its osmo-protectant action, and microbial consortia, used to increase soil biological fertility and to promote plant growth [47], showed some positive effects on growth rate trend (Figure 1), but even they did not prevent KiDS onset (Table 1). The root system of plants inoculated with the microbial consortium 1 (TiMi1) at the end of the trial was the best-developed and rich of fibrous roots (Figure S5). Glycine-betaine produced the best performance when associated to microbial consortium 2 (Figure 1).

A new trial is underway to check the efficacy of climate-controlling orchard management, with over-tree micro-sprinkler irrigation and crop protection with different covering materials.

## 5. Conclusions

The agronomic treatments tested in the experimental orchard, aimed at improving plant wellness, were not effective in preventing KiDS occurrence, even if some of them resulted in positive effects. A negative effect of composting can be signaled, while the

inoculum of selected microbial consortia in soil and the leaves protection from desiccation with glycine-betaine as an osmo-protectant can have positive effects. The physiological behavior of plants and the anatomical features observed in roots xylem suggest a probable role of weather in the KiDS onset: in particular, high VPD, a well-known stress factor inducing plants adaptation strategies, could be very detrimental for kiwifruit, recognized as an anisohydric plant. Then, adaptation to climate change in Mediterranean area could be the main cause of KiDS occurrence, even if KiDS multifactor nature can be considered confirmed. The adaptation of kiwifruit to long-term climatic variations can make it more sensitive to sudden and extreme weather events, such as very high, fast rising temperature and VPD that happen more and more frequently in summer causing strong and abrupt water stress. Considering the endemic presence of the bacterial disease induced by PSA in the same area, we postulate a possible synergistic negative effect of bacterial infection and climate stress: this could also explain the presence of healthy and KiDS-affected orchards in very close places, apparently facing the same climatic stresses.

Giving the multifactor nature of KiDS, several further attempts can still be made to promote a more effective resilience of this crop to climate change by agronomic management focused and targeted to orchard climate control, based on the experiences already gained in good soil and water management practices. The interplay between climate change and biotic stresses induced by endemic pathogens should also be considered and studied with more attention.

**Supplementary Materials:** The following supporting information can be downloaded at: https://www.mdpi.com/article/10.3390/horticulturae8100906/s1, Figure S1: kiwifruit cv Hayward. A: plants used for plantation. B: plant from the experimental orchard at the second year of growth; Figure S2: field where the experimental orchard was planted (Saluzzo, CN, 44°38′49.8″ N 7°31′52.6″ E); Figure S3: KiDS symptoms found in Saluzzo region. A: leaves chlorosis and desiccation, petiole wilt. B: roots from a healthy (left) and from a KiDS affected (right) plant; Figure S4: typical early bacterial disease PSA symptoms (canker, cortical necrosis, exudates); Figure S5: entire root systems of plants explanted at the end of the trial (end of the third year).

**Author Contributions:** Conceptualization, L.B. and C.M. (Chiara Morone); writing—original draft preparation, L.B.; writing—review and editing, L.B., L.N., C.M. (Chiara Morone), C.M. (Claudio Mandalà); SEM analysis, M.G.F.; LM analysis, C.M. (Claudio Mandalà); methodology, L.N. and C.M. (Claudio Mandalà); formal analysis, L.B. and L.N.; visualization, L.B. and C.M. (Claudio Mandalà); investigation, L.N., M.S. and C.A.M.; project administration, C.M. (Chiara Morone). All authors have read and agreed to the published version of the manuscript.

**Funding:** This research was funded by Regione Piemonte, Direzione Agricoltura, Settore Servizi di sviluppo e controlli per l'Agricoltura, Determinazione n. 1066 del 30 October 2017, Programma Regionale Ricerca e Sperimentazione Agricola 2017–2019 and The APC was offered by MDPI Horticulture. The funders had no role in the design of the study; in the collection, analyses, or interpretation of data; in the writing of the manuscript, or in the decision to publish the results.

**Data Availability Statement:** Not applicable.

**Acknowledgments:** The authors acknowledge and remember with great esteem and respect Giuliano Sacchetto and son, for providing its own orchard for experimentation and for managing it with great accuracy and reliability; Giovanna Cressano for flawless administrative support and for her willingness and helpfulness; CCS Aosta s.r.l. and Green Ravenna s.r.l. for providing microbial consortia and related commercial product; Marta Vallino for kind readiness to help with her skills and instruments for rotary microtome sections preparation for LM analysis; and the nice colleague Giancarlo Peiretti for bothering the perfectly healthy kiwifruit plant of his private garden, taking samples of roots sacrificed for the research.

**Conflicts of Interest:** The authors declare no conflict of interest.

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
