# Peer review of "Kiwifruit Adaptation to Rising Vapor Pressure Deficit Increases the Risk of Kiwifruit Decline Syndrome Occurrence"

_horticulturae, doi:10.3390/horticulturae8100906_

Round 1

Reviewer 1 Report

Dear Authors and Editors
The article was written in a transparent manner. In its assumptions, the study is justified and interesting. Similar research has not been done so far.
The topics presented in the article are appropriate for the Journal's profile.
The authors presented the discussed issues in a broad way.
The literature is selected in the right way.
The conclusions sum up the entire article appropriately.
In my opinion, this paper can be accepted for publication in
Horticulturae. I found only very minor errors in the manuscript:

- page 9 line 357 after FlCor. space

- in the reference list in point 15, 20, 30, 31, 38 change the font color from blue to black

Best regards

Author Response

Author's Reply to the Review Report (Reviewer 1)

The article was written in a transparent manner. In its assumptions, the study is justified and interesting. Similar research has not been done so far.
The topics presented in the article are appropriate for the Journal's profile.
The authors presented the discussed issues in a broad way.
The literature is selected in the right way.
The conclusions sum up the entire article appropriately.
In my opinion, this paper can be accepted for publication in
Horticulturae.

The authors are thankful for the reviewers´ comments

I found only very minor errors in the manuscript:

- page 9 line 357 after FlCor. Space

Done

- in the reference list in point 15, 20, 30, 31, 38 change the font color from blue to black

Done

Reviewer 2 Report

The manuscript by Bardi et al. intended to discover the cause of KiDS related to climate change, i.e., global warming, and find the strategies to face KiDS occurrence in crops. It is an interesting topic and is meaningful for the kiwifruit cultivation. However, there are some drawbacks about the introduction section, experimental design, results’ analysis and graphics preparation. My major concerns are list as following:

1、  The title is too vague. The climate change should be defined in detail, in what aspect? Besides, I feel that it is contradictory by simultaneously present the phrases “adaptation” and “sensitive” in the title, this make me confused.

2、  In the Abstract part, the manuscript should simply introduce the experimental design. The reason why the VPD was investigated is also not stated. Line 25, what is the mean of “agronomical practices”? And what is the Anatomical and physiological adaptations in Line 26? The Abstract part need to be revised, and the main results in detail should be stated. In fact, after reading the Abstract part, I am still not clear what is the suggested solution solving the KiDS issue.

3、  Line 79-80, the manuscript states that the climatic factors are probably contributing to determine the onset and the worsening of KiDS. The authors need to cite more literature(s) that was (were) published by other researchers to support this viewpoint. Besides, the manuscript focused on the relationship between climate change and KiDS, but the literatures summary about this relationship in the Introduction part is inadequate and is lack of logic.

4、  How could the authors prove that the growth, health, and physiological behavior of the plants were related to the environmental conditions connected to climate change just by the setting up several different agronomical practices. I doubt that the effect of climate change on the KiDS can be investigated just by implementing several different agronomical practices experiments and synchronously monitoring the environmental conditions.

5、  Line 166, the unit of WUE is wrongly written.

6、  Those figures in this manuscript should be re-prepared, i.e., delete the horizontal line, showed the y-axis, the legend “Stem growth” is not necessary. Table 1 should be drawn as a three-line table.

7、  Line 177, why the leaf water potential was measured at Midday (from 12.00 to 15.00) but not the same time as gas exchange measurement?

8、  Line 285-286, the statement is contradictory. However, the letters above the bar in Figure 3 showed that there was significant difference between the values in some treatments.

9、  Line 309-310, the manuscript stated that the detected values were high, but what was the normal value of the WUE of kiwifruit plants?

10、              The measures of leaf gas exchange were carried out during the second and third year from plantation, in mid-June (20 days after mid bloom, DAMB), mid-July (50 DAMB) and mid-August (80 DAMB), it seemed there should be at least 3 groups of data, but the Figure 3 only showed the data obtained in mid-July (50 DAMB), why? How about the other data? And the leaf water potential also showed only the data obtained in mid-June. I believe that the continuous monitoring of leaf gas exchange during the second and third year from plantation is necessary, even if one time per month is workable, since the manuscript intended to investigate the relationship between KiDS and climate change, and the climate change is a long-term effect and complex.

11、              In addition, the measurements of the growth and physiological behavior of the plants should also be conducted at the beginning and the end of the whole experiment period (three years), in order to better understand the climate change effect on those parameters performance.

12、              Some statements about the cause of KiDS in the Discussion part (i.e., the first three paragraphs) can be moved to the Introduction part to make the introduction more informative.

13、Line 445-446, the sentence stated that water stress was correlated with high leaf water potential, is this true? I believe that low water potential always occurs under water stress conditions.

Author Response

Author's Reply to the Review Report (Reviewer 2) 

The manuscript by Bardi et al. intended to discover the cause of KiDS related to climate change, i.e., global warming and find the strategies to face KiDS occurrence in crops. It is an interesting topic and is meaningful for the kiwifruit cultivation. However, there are some drawbacks about the introduction section, experimental design, results’ analysis and graphics preparation.

The authors are thankful for the reviewers´ comments because they allow to significantly improve the quality of our work

My major concerns are list as following:

1The title is too vague. The climate change should be defined in detail, in what aspect? Besides, I feel that it is contradictory by simultaneously present the phrases “adaptation” and “sensitive” in the title, this make me confused.

This comment is very useful. We propose the following alternative title: “Kiwifruit adaptation to rising vapor pressure deficit increases the risk of kiwifruit decline syndrome occurrence”

2 In the Abstract part, the manuscript should simply introduce the experimental design. The reason why the VPD was investigated is also not stated. Line 25, what is the mean of “agronomical practices”? And what is the Anatomical and physiological adaptations in Line 26? The Abstract part need to be revised, and the main results in detail should be stated. In fact, after reading the Abstract part, I am still not clear what is the suggested solution solving the KiDS issue.

The abstract has been completely revised according to these very useful suggestions.

3、 Line 79-80, the manuscript states that the climatic factors are probably contributing to determine the onset and the worsening of KiDS. The authors need to cite more literature(s) that was (were) published by other researchers to support this viewpoint. Besides, the manuscript focused on the relationship between climate change and KiDS, but the literatures summary about this relationship in the Introduction part is inadequate and is lack of logic.

and

12 Some statements about the cause of KiDS in the Discussion part (i.e., the first three paragraphs) can be moved to the Introduction part to make the introduction more informative.

As far as we know, the relationship between climate change and KiDS was not directly investigated in the past; this possible relationship was suggested for the first time in 2020, basing on the current knowledge on other plants, as stated in Introduction, lines 39-49. In order to make the Introduction more explanatory of the current knowledge on kiwifruit and on the possible relationship between climate change and KiDS, we moved to Introduction the first three paragraphs of Discussion, as rightly suggested in Comment n. 12.

4How could the authors prove that the growth, health, and physiological behavior of the plants were related to the environmental conditions connected to climate change just by the setting up several different agronomical practices. I doubt that the effect of climate change on the KiDS can be investigated just by implementing several different agronomical practices experiments and synchronously monitoring the environmental conditions.

This statement is perfectly true. It is really difficult to carry out a crop trial to fully investigate the effects of climate change on orchards. As KiDS is a great, actual problem whose causes are still not completely explained, our aim was to look for “possible practices effective to prevent severe plant damages”,  and at the same time “we observed the behavior of kiwifruit plants in an experimental kiwifruit orchard” in order to improve our knowledge on the physiology and anatomy of this plant by exploring the possible effects induced by environmental stresses, basing on the theoretical knowledge and on the weather data detected in the experimental crop.  Our findings support the hypothesis based on the possible effects of rising VPD, and the umpteenth lack of significant effects from different agronomic managements could be considered a further evidence that the origin of KiDS should be seeked elsewhere. As the results reached were much more interesting for the possible effects of VPD than of agronomic management, we emphasized that.  We significantly modified the description of experimental trial, in order to explain more clearly the setting and to avoid misunderstandings; moreover, we enhanced the description and the discussion of  results and reformulated significantly the conclusions to improve the manuscript accounting for this very important comment.

5Line 166, the unit of WUE is wrongly written.

Corrected

6Those figures in this manuscript should be re-prepared, i.e., delete the horizontal line, showed the y-axis, the legend “Stem growth” is not necessary. Table 1 should be drawn as a three-line table.

The figures and table were edited following these suggestions

7Line 177, why the leaf water potential was measured at Midday (from 12.00 to 15.00) but not the same time as gas exchange measurement?

Midday leaf water potential was chosen as an index of the plant water status because it is considered the most representative index when the VPD-induced water stress is evaluated, instead of pre-down leaf water potential, that is related to soil water potential. Gas exchanges were measured in the morning, when stomatal conductance and the other related parameters are highest, in order to evaluate possible differences among treatments (gas exchanges drop drastically in kiwifruit during the middle of the day)

8Line 285-286, the statement is contradictory. However, the letters above the bar in Figure 3 showed that there was significant difference between the values in some treatments.

The sentence was rephrased

9Line 309-310, the manuscript stated that the detected values were high, but what was the normal value of the WUE of kiwifruit plants?

There are few data available in literature on WUE of kiwifruit. As Figure 4 was modified according to comment n. 11, we added a comment on WUE increase from second to third year; moreover, we added a reference to a recent paper concerning physiology (including WUE) of kiwifruit under water deficit.

10、              The measures of leaf gas exchange were carried out during the second and third year from plantation, in mid-June (20 days after mid bloom, DAMB), mid-July (50 DAMB) and mid-August (80 DAMB), it seemed there should be at least 3 groups of data, but the Figure 3 only showed the data obtained in mid-July (50 DAMB), why? How about the other data? And the leaf water potential also showed only the data obtained in mid-June. I believe that the continuous monitoring of leaf gas exchange during the second and third year from plantation is necessary, even if one time per month is workable, since the manuscript intended to investigate the relationship between KiDS and climate change, and the climate change is a long-term effect and complex.

This comment is really useful because it allowed us to realize that the description of the activities was not accurate.  In fact, we measured leaf gas exchanges three times during the second year, but just one time during the third year, in July (whose results are shown in Figure 3): we made this decision because the results of the first year suggested that July was the best moment to evidence differences among treatments, and July was also the time in which plants started showing signs of stress due to high temperatures. We decided not to show all the collected data to avoid an excess of figures in the manuscript, focusing only on the more significant data. We revised both Materials and methods and Results to better explain that. We could add a table with complete data of leaf gas exchanges as Supplementary material if it is considered necessary.  Regarding leaf water potential, unfortunately the data of July are uncomplete, but we considered strongly interesting, then worthy of being shown, that so low values were detected already in June, when the plants did not show severe stress symptoms yet. We have added a sentence about the increase observed in July for some of the treatments for which the data are available.

We are fully aware that a work on the effects of climate change should be carried out differently, but as we stated in the answer to comment 4., our initial goal was to look for agronomic solutions to KiDS, but as our results were much more interesting for the possible effects of VPD than of agronomic management, we emphasized that. We have now ongoing a project in which we are monitoring KiDS occurrence in the whole national territory in relation to the trend of temperature and VPD during the last 10 years: this project was started also owing to the results that we are showing in the present manuscript, that we wish could be considered suitable for publication.

11              In addition, the measurements of the growth and physiological behavior of the plants should also be conducted at the beginning and the end of the whole experiment period (three years), in order to better understand the climate change effect on those parameters performance.

During the first year it was not possible to carry out measurements because of the very small size of the plants. In the revised manuscript we added comments on the evolution from the second to the third year regarding stem growth, leaf gas exchanges and water use efficiency. Figure 1 and Figure 4 were modified by adding the data of the second year, evidencing the decrease of growth rate and the increase of WUE from second to third year. Finally, we consider very significant the data reported in Figure 5, that is representative of the whole experiment period, and is referred to VPD measured directly in real time in the experimental orchard.

13、Line 445-446, the sentence stated that water stress was correlated with high leaf water potential, is this true? I believe that low water potential always occurs under water stress conditions.

Sorry for the mistake; it was corrected (low water potential)

Reviewer 3 Report

Some corrections that need to be done:

In Abstract:  Line 18: please delte 'whose nature is commonly recognized as non-pathogenic' -->You mentioned here that this syndrome is non-pathogenic; when we talk about pathogenecity is when it is due to fungus/bacterial whearas this sydrome is mainly due to environmental factors. 

line 22: to put 'origin of this syndrome' instead of 'origin'

In Introduction: line 95: addition of  zeolite to soil: add a sentence explaining why you used zeolite ( to improve what in the soil ??) Same in Line 96: Addition of glycine-betaine to leaves why? (to......)

In Materials and Methods: Line 98: the title 2.1 should be in itallic and capital letters for the 1st words you have to chek this in all the document ( see author instruction).

Lines 101 and 102: add Figures: 1) of the field where you conducted your experiment, 2) of the cv Hayward, 3) also a pictures of the KiDS symptoms you found in Saluzzo region.

Line 106: You have to mention the name of the experimental design you used.  it is 'Randomized Block Design' is that correct? You have 4 blocks and in each block 11 treatments. How many plants per treatment you used; you have to add the detail. and also add a scheme with the different bloks and the different treatment in each block. This will enhance the description you made.

Line 132: for the compost you have to add when you put it (before planting ?? or what)

Line 162: I Suggest the following:

2.2. Evaluated Parameters

2.2.1. Leaf Gas Exchange

2.2.2. Leaf Water Potential

2.2.3. Stem Growth

2.2.4. KiDS and Bacterial Disease PSA Symptoms

and so on ......

line 165: 'were' istead of 'where'

Line 181: the subtitle 2.4 I suggest to separete in different parameters and be specific (see my comments on line 162) as follows:

2.2.3. Stem Growth

2.2.4. KiDS and Bacterial Disease PSA Symptoms

Lines 183 and 191: chek it is '40' or '-40'. Besides in this paragraph it is important to add pictures of the different symptoms caused by the bcterial disease PSA and those of KiDS. This will help readers to see differences !!!

In Results section:

Line 252: put 'the bacterial disease PSA' instead of 'PSA'. In all the document when you talk about PSA be specific and mention that.

In figure 3 section 3.4 (the group of 4 parameters) page 7: you should add the standard deviation bar on each treatment.

Figure 4 in page 8: you should also add standard deviation or standar error bars on each treatment 

Line 329: you have to put 'stomatal conductance (gs, mol H2O m-2 s-1)' 

Line 332: add a sentence of the most relevant result from this analisis you made.

Figure 5: Line 335: Add 'vapor pressure deficit, VPD (kPa)'

In the title of figures, you should put the hole words of the abreviation.

In Figure 6 Page 10: If you have pictures of the entire roots add them, in order to see the differences in the entire roots also. Besides in the pictures a, b,c and d you might add arrows to show the xylem vessels. In the title of this figure 6 you should add at the end in line 369: Differences between treatments mainly in the number and diameter of xylem vessels.

- In this Figure 6, why you did not present all the 11 treatments ? it would be interessting if you add all of them.  

In the title of Fig 6. Line 367: in a: delete from a private garden and put  

a = root from a perfectly healthy kiwifruit plant.

Figure 8, line 391: Add 'scanning electron microscope' (SEM).

Figure 9, Line 408: Add 'vapor pressure deficit, VPD (kPa)'

Discussion section: Some sentences need to be reformulated and english corrected. 

Author Response

Author's Reply to the Review Report (Reviewer 3)

Some corrections that need to be done:

In Abstract:  Line 18: please delte 'whose nature is commonly recognized as non-pathogenic' -->You mentioned here that this syndrome is non-pathogenic; when we talk about pathogenecity is when it is due to fungus/bacterial whearas this sydrome is mainly due to environmental factors

The sentence has been deleted.

line 22: to put 'origin of this syndrome' instead of 'origin'

The addition has been done.

In Introduction: line 95: addition of  zeolite to soil: add a sentence explaining why you used zeolite ( to improve what in the soil ??) Same in Line 96: Addition of glycine-betaine to leaves why? (to......)

The sentences have been added.

In Materials and Methods: Line 98: the title 2.1 should be in itallic and capital letters for the 1st words you have to chek this in all the document ( see author instruction).

The text has been edited as suggested.

Lines 101 and 102: add Figures: 1) of the field where you conducted your experiment, 2) of the cv Hayward, 3) also a pictures of the KiDS symptoms you found in Saluzzo region.

Pictures were added as suggested as Supplementary material

Line 106: You have to mention the name of the experimental design you used.  it is 'Randomized Block Design' is that correct? You have 4 blocks and in each block 11 treatments. How many plants per treatment you used; you have to add the detail. and also add a scheme with the different bloks and the different treatment in each block. This will enhance the description you made.

The experimental design was a block design, but I’m not sure I can define it “randomized”: in fact, the orchard was organized in rows; each row corresponded to a treatment; the blocks were designed perpendicularly to rows: so, each row was divided in four blocks. The plants used for parameters evaluations were  selected randomly inside each block, in order to analyse at least four plant per treatment. You must consider also that each block was formed by about 20-25 plants per row. We have added a more detailed description.

Line 132: for the compost you have to add when you put it (before planting ?? or what)

The sentence (added before planting) has been added.

Line 162: I Suggest the following:

2.2. Evaluated Parameters

2.2.1. Leaf Gas Exchange

2.2.2. Leaf Water Potential

2.2.3. Stem Growth

2.2.4. KiDS and Bacterial Disease PSA Symptoms

and so on ......

The text has been edited as suggested

line 165: 'were' istead of 'where'

Done

Line 181: the subtitle 2.4 I suggest to separete in different parameters and be specific (see my comments on line 162) as follows:

2.2.3. Stem Growth

2.2.4. KiDS and Bacterial Disease PSA Symptoms

Done

Lines 183 and 191: chek it is '40' or '-40'.

It is -40 (40 days before mid bloom= -40 DAMB (days after mid bloom))

Besides in this paragraph it is important to add pictures of the different symptoms caused by the bcterial disease PSA and those of KiDS. This will help readers to see differences !!!

Figure S3 and S4 have been added

In Results section:

Line 252: put 'the bacterial disease PSA' instead of 'PSA'. In all the document when you talk about PSA be specific and mention that.

Done

In figure 3 section 3.4 (the group of 4 parameters) page 7: you should add the standard deviation bar on each treatment.

Done

Figure 4 in page 8: you should also add standard deviation or standar error bars on each treatment 

Done. Data of second year measurements were also added (as suggested by Reviewer 3)

Line 329: you have to put 'stomatal conductance (gs, mol H2O m-2 s-1)' 

Done

Line 332: add a sentence of the most relevant result from this analisis you made.

Done

Figure 5: Line 335: Add 'vapor pressure deficit, VPD (kPa)'

Done

In the title of figures, you should put the hole words of the abreviation.

Figures editing was modified according to suggestions from Reviewer 2; titles were eliminated where not necessary (so also abbreviations)

In Figure 6 Page 10: If you have pictures of the entire roots add them, in order to see the differences in the entire roots also. 

Figure S5 has been added, and several related new comment in the text (after Figure 6 and in relation to Figure 1 – growth rate)

Besides in the pictures a, b,c and d you might add arrows to show the xylem vessels. In the title of this figure 6 you should add at the end in line 369: Differences between treatments mainly in the number and diameter of xylem vessels.

Done

- In this Figure 6, why you did not present all the 11 treatments ? it would be interessting if you add all of them.  

Unfortunately the quality of our pictures was not homogeneous. We have chosen the best and more representative pictures

In the title of Fig 6. Line 367: in a: delete from a private garden and put  

a = root from a perfectly healthy kiwifruit plant.

Done

Figure 8, line 391: Add 'scanning electron microscope' (SEM).

Done

Figure 9, Line 408: Add 'vapor pressure deficit, VPD (kPa)'

Done

Discussion section: Some sentences need to be reformulated and english corrected

We revised the whole text; we wish we have improved it adequately

Round 2

Reviewer 2 Report

In Figure 1 and 4, the lowercase letters showed be added to show the significant differences between the treatments.

Author Response

In Figure 1 and 4, the lowercase letters showed be added to show the significant differences between the treatments.

We acknowledge the reviewer for the suggestions that allowed us to significantly improve our manuscript.

We have added the lowercase letters to Figures 1 and 4 to show the results of the ANOVA analysis.